# MULTI-AGENT MULTI-GAME ENTITY TRANSFORMER

## ABSTRACT

Building large-scale generalist pre-trained models for many tasks is becoming an emerging and potential direction in reinforcement learning (RL). Research such as Gato and Multi-Game Decision Transformer have displayed outstanding performance and generalization capabilities on many games and domains. However, there exists a research blank about developing highly capable and generalist models in multi-agent RL (MARL), which can substantially accelerate progress towards general AI. To fill this gap, we propose Multi-Agent multi-Game ENtity TrAnsformer (MAGENTA) from the entity perspective as an orthogonal research to previous time-sequential modeling. Specifically, to deal with different state/observation spaces in different games, we analogize games as languages by aligning one single game to one single language, thus training different "tokenizers" and a shared transformer for various games. The feature inputs are split according to different entities and tokenized in the same continuous space. Then, two types of transformer-based model are proposed as permutation-invariant architectures to deal with various numbers of entities and capture the attention over different entities. MAGENTA is trained on Honor of Kings, Starcraft II micromanagement, and Neural MMO with a single set of transformer weights. Extensive experiments show that MAGENTA can play games across various categories with arbitrary numbers of agents and increase the efficiency of fine-tuning in new games and scenarios by 50%-100%. See our project page at https://sites.google.com/view/rl-magenta.

## 1 INTRODUCTION

In recent years, transformer-based models, as a solution to build large-scale generalist models, have made substantial progress in natural language processing (Brown et al., 2020; Devlin et al., 2018), computer vision (Dosovitskiy et al., 2020; Bao et al., 2021), graph learning Yun et al. (2019); Rong et al. (2020), Furthermore, they are showing their potential in reinforcement learning (RL) (Reed et al., 2022; Lee et al., 2022; Wen et al., 2022) by modeling and solving sequential problems.

However, there are few inherent challenges in building large-scale general RL agents. First, when the training environment and the test environment are the same, RL is inclined to overfit the training environments, while lacking generalizability to unknown environments. As a result, a model is often needed to be re-trained from scratch for a new task. Second, it is challenging for a single model to adapt for various environments with differences in the numbers of agents, states, observations, actions, and dynamics. Third, training from scratch normally suffers from expensive computational cost, especially for large-scale RL. For example, AlphaStar requires 16 TPUs to train for 14 days, and Honor of King (HoK) requires 19,600 core CPUs and 168 V100 GPUs to train for nearly half a month. Thus, building a general, reusable, and efficient RL model has been an increasingly important task for both industrial and non-industrial research.

To this end, we investigate whether a single model, with a single set of parameters, can be trained by playing multiple multi-agent games in an online manner, which is a blank in current research after Gato (Reed et al., 2022) and MGDT (Lee et al., 2022). We consider training on Honor of Kings (HoK), Starcraft II micromanagement (SMAC), and Neural MMO (NMMO), informally asking:

*Can models learn some general knowledge about games across various categories?*

In this paper, we answer this question by proposing Multi-Agent multi-Game ENtity TrAnsformer (MAGENTA). We consider this problem as a few-shot transfer learning with the hypothesis, where a

single model is capable of playing many games and can be adapted to never-seen-before games or scenarios with fine-tuning. For interpretability, we align the design of MAGENTA with the Entity Component System (ECS) architectural pattern of video games and the multilingual transformer. Specifically, we treat different games as different languages by aligning one single game to one single language as shown in Fig.3. Different languages have different tokenizers, so do games. Also, we expect that the learned representations in different games are similar to the word2vec in NLP. In this case, our transformer can be viewed as a multilingual transformer to capture the common knowledge among different games. We split the feature input according to different entities and tokenize the features in the same continuous space. Unlike existing applications of the transformer as a causal time-sequential model, the transformer in MAGENTA serves as a permutation-invariant architecture to attend to the features of different entities. We propose two types of transformer architecture, Encoder-Pooling (EP) and Encoder-Decoder (ED), to build permutation-invariant models. In this way, the output of the model should not change under any permutation of the elements in the input. Lastly, we show the training scheme of MAGENTA about how to train and transfer the pre-trained model.

As a step towards developing a general model in RL/MARL, our contributions are threefold: First, by aligning games and languages, we show that it is possible to train a single generalist agent to act across multiple environments in an online manner. Second, we present permutation-invariant models for adapting various numbers of agents in different games. And third, we find that MAGENTA can achieve rapid fine-tuning in an online fashion to different scenarios within a single game, to a new type of game, and to different numbers of agents. Furthermore, we release the pre-trained models and code to encourage further research in this direction.

## 2 PRELIMINARY

### 2.1 REINFORCEMENT LEARNING AND MULTI-AGENT RL

An RL problem is generally studied as a Markov decision process (MDP) (Bellman, 1957), defined by the tuple: $\text{MDP} = (\mathcal{S}, \mathcal{A}, \mathcal{P}, r, \gamma, T)$, where $\mathcal{S} \subseteq \mathbb{R}^n$ is an $n$-dimensional state space, $\mathcal{A} \subseteq \mathbb{R}^m$ an $m$-dimensional action space, $\mathcal{P} : \mathcal{S} \times \mathcal{A} \times \mathcal{S} \to \mathbb{R}_+$ a transition probability function, $r : \mathcal{S} \to \mathbb{R}$ a bounded reward function, $\gamma \in (0, 1]$ a discount factor and $T$ a time horizon. In MDP, an agent receives the current state $s_t \in \mathcal{S}$ from the environment and performs an action $a_t \in \mathcal{A}$ defined by a policy $\pi_\theta : S \to \mathcal{A}$ parameterized by $\theta$. The objective of the agent is to learn an optimal policy: $\pi_{\theta^*} := \text{argmax}_{\pi_\theta} \mathbb{E}_{\pi_\theta} \left[ \sum_{i=0}^{T} \gamma^i r_{t+i} | s_t = s \right]$. An MARL problem is formulated as a *decentralised partially observable Markov decision process* (Dec-POMDP) (Bernstein et al., 2002), which is described as a tuple $\langle n, \boldsymbol{S}, \boldsymbol{A}, P, R, \boldsymbol{O}, \boldsymbol{\Omega}, \gamma \rangle$, where $n$ represents the number of agents. $\boldsymbol{S}, \boldsymbol{A}, P, R$ are all global versions of those in MDP. $\boldsymbol{O} = \{O_i\}_{i=1,\cdots,n}$ denotes the space of observations of all agents. Each agent $i$ receives a private observation $o_i \in O_i$ according to the observation function $\boldsymbol{\Omega}(s, i) : \boldsymbol{S} \to O_i$. [1].

### 2.2 ATTENTION MECHANISM IN TRANSFORMER

One of the most essential components of Transformer (Vaswani et al., 2017) is the attention mechanism, which captures the interrelationship of input sequences. Assume that we have $n$ query vectors (corresponding to a set with $n$ elements) each with dimension $d_q : Q \in \mathbb{R}^{n \times d_q}$. The attention function is written as $\text{Attention}(Q, K, V) = \omega(QK^\top)V$, which maps queries $Q$ to outputs using $n_v$ key-value pairs $K \in \mathbb{R}^{n_v \times d_q}, V \in \mathbb{R}^{n_v \times d_v}$. The pairwise dot product $QK^\top \in \mathbb{R}^{n \times n_v}$ measures how similar each pair of query and key vectors is, with weights computed with an activation function $\omega$. The output $\omega\left(QK^\top\right)V \in \mathbb{R}^{n \times d_v}$ is a weighted sum of $V$ where a value gains more weight if its corresponding key has a larger dot product with the query. Furthermore, self-attention refers to cases where $Q, K, V$ share the same set of parameters. Multi-head attention is an extension of the attention by computing $h$ attention functions simultaneously and outputting a linear transformation of the concatenation of all attention outputs.

### 2.3 ENTITY IN VIDEO GAME DESIGN

In video game development, the Entity Component System (ECS) is the often used software architectural pattern for the representation of game world objects (Bilas, 2002). An ECS has three parts:

---

[1]We present this paper in the scope of MARL, and abuse the term "feature" an alternative of observation.

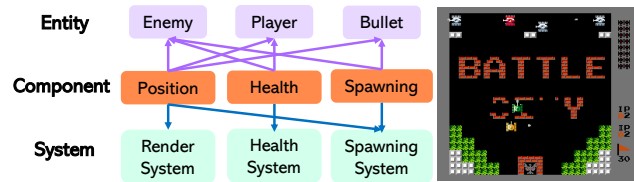

Figure 1: An Example of ECS of the BattleCity game in Atari 2600.

entities, components, and systems. Entities are composed of data from components, while systems operate on entities' components. For example as shown in Fig.1, in BattleCity, the enemy tank, the player, and the bullets can be represented as different entities. These entities that can cause damage might have a health component. The physics system in this game may query entities whether having mass, velocity, as well position components, and do physics calculations on the sets of components for each entity.

## 3 RELATED WORK

### 3.1 TRANSFORMER IN RL

Recently, transformer-based models have been shown to be a powerful tool for RL to capture global dependencies or model long-term squence. The architecture of MAGENTA is closely related to AlphaStar (Jaderberg et al., 2019) and OpenAI Five (OpenAI, 2019), which use transformer-based models to embed unit features for each player. Another perspective of using transformer-based models is to model an RL problem as a sequence prediction problem that models trajectories autoregressively, such as Decision Transformer (DT) (Chen et al., 2021), Trajectory Transformer (TT) (Janner et al., 2021), GATO (Reed et al., 2022), and Multi-Game DT (MGDT) (Lee et al., 2022). In addition, Multi-Agent transformer (MAT) (Wen et al., 2022) considers the MARL problem as a sequence prediction problem that generates the optimal action of each agent sequentially.

However, these transformer-based models study the generality with *trajectory sequences* in *single agent* scenarios (Reed et al., 2022; Lee et al., 2022) or focus on *a single task* with the sequences (Jaderberg et al., 2019; OpenAI, 2019; Chen et al., 2021; Janner et al., 2021; Wen et al., 2022). In addition, they use causal transformer decoders, which use causal masking to discard future information. MAGENTA fills the current research blank by using a transformer model without positional encoding and exploring generality with entity sequences in multi-agent scenarios. Note that models such as DT, TT, and MGDT utilize the sequence along the time dimension, while MAGENTA along the entity dimension, so MAGENTA is orthogonal to time trajectory sequence modeling methods by viewing MAGENTA as a tokenizer in these methods. We summarize these related works as shown in Table 1.

Table 1: Related Work on Transformers in RL

| Method | Sequence | Data | Agent | Task | Transfer | Games/Domains |
|---|---|---|---|---|---|---|
| DT, TT | trajectory | offline | single | single | No | Atari,OpenAI Gym,Key-to-Door |
| Gato | trajectory | offline | single | multi | No | Atari,caption,chat,robotics,etc. |
| MGDT | trajectory | offline | single | multi | Yes | Atari |
| online DT | trajectory | online | single | single | Yes | OpenAI Gym |
| MAT | agent | online | multi | single | No | Starcraft II micromanagement |
| AlphaStar | entity | online | multi | single | No | Starcraft II full game |
| OpenAI Five | entity | online | multi | single | No | Dota 2 |
| MAGENTA | entity | online | multi | multi | Yes | HoK,Neural MMO, Starcraft II micromanagement |

### 3.2 REUSE TRAINED RL MODELS

Another related research direction is to reuse trained RL models, which is an important subfield of RL. Some approaches directly use the trained models. For example, to avoid restarting from scratch after changes in code and environment, OpenAI Five (OpenAI, 2019) used the "surgery" approach to convert a trained model to certain larger architectures with customized weight initialization.

Figure 2: (Left) Overall framework of MAGENTA. (Right) An example tokenizer for HoK. Spatial features are as images and entity features as vectors.

MGDT (Lee et al., 2022) and Gato (Reed et al., 2022) follow the manner of few-shot transfer learning by using a single set of weights to play many games or handle tasks in multiple domains simultaneously. Furthermore, some approaches implicitly exploit trained models, such as behavior cloning (Gao et al., 2018; Hester et al., 2018), distillation (Rusu et al., 2015; Parisotto et al., 2015), and the use of the teacher-student framework (Chang et al., 2015; Schmitt et al., 2018; Uchendu et al., 2022). Considering that all these methods are based on well-pretrained model weights, MAGENTA follows the fashion of direct usage and provides our code and trained agents. Interested researchers can use the provided models for further exploration on how to reuse such models.

## 4 MULTI-AGENT MULTI-GAME ENTITY TRANSFORMER

In building generic multi-agent AI for different games, the following challenges are encountered: (1) different feature inputs (i.e., different observations), different action spaces, and different rewards across different classes of games; (2) the number of entities varies across different classes of games.

Motivated by multilingual models (Devlin et al., 2018; Liu et al., 2020; Xue et al., 2021) in the NLP community, we treat different games as different languages to deal with different inputs of features. Although words or sentences in different languages have different representations, such as "*I love you*" in English, and "*Je t'aime*" in French; however, the meaning behind those words is the same. We also find the analogy in the ECS game design pattern. Games are designed based on physical rules in the world. Basically, entities in most games can move, collide, and even attack. The components or attributes of these entities are changing according to the "virtual rules", which are mimics of the physical rules. The RL policy can be considered as a neural network system to approximate such rules. So we divide such neural network system into three parts: tokenizor, transformer, and output. Tokenziors are trained for different games, just as they are trained for different languages in NLP. Transformer is a multi-game transformer, as an analogy to the multilingual transformer. While output are finetuned according to the down-streaming tasks.

### 4.1 GAMES AS LANGUAGES

Tokenizor is used to obtain initial embeddings of words in the NLP community. When we treat games as languages, different features of entities can be analogized to different "words". Such "words" are typically composed of images and vectors. Therefore, we need to divide the features into "words" and then map them into the same *continuous* space.

A good tokenization for different games should be feature-space-agnostic. Different games have different features, such as images or vectors. Furthermore, the features of different entities have different shapes, dimensions, and meanings. Therefore, a good tokenization is supposed to handle different feature spaces and split the entire feature into individual elements, such as the tokenizer in NLP splits sentences into words.

We categorize the feature input into two types: images and vectors. When playing games, we can have a screenshot as an image. Typically, we call them frames as shown in (Mnih et al., 2015). The numerical attributes of the entities can be vectors. For example, a vector composed of the health point, the mana point, the speed, the attack, and defense, etc. As shown in Fig.2, to process images, we first use convolution neural networks to extract the latent representation and then linearly project the representation into the embedding space. For processing vectors, we divide the vectors into many inputs that correspond to different entities. Then we use a fully connected neural network to get tokens for different entities. The same types of entities share the same parameters in neural networks.

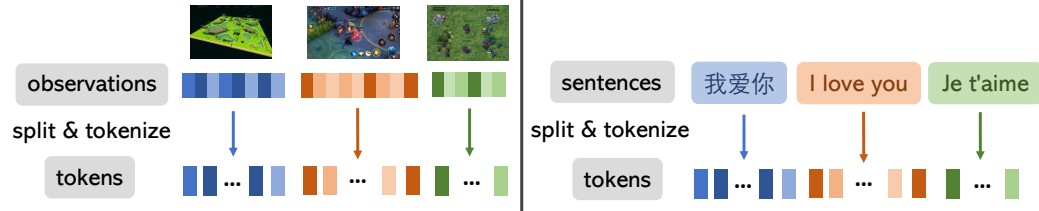

Figure 3: Games as Languages. (Left) Tokenize different observations from different games. (Right) Tokenize different sentences from different languages.

We chose our tokenization scheme with simplicity in mind, but many other schemes are possible. For example, more sophisticated methods, such as VAE (Kingma & Welling, 2013) can be used to learn a more effective token representation.

### 4.2 TRANSFORMER IN MAGENTA

In many game scenarios, the number of entities to be modeled changes over time (e.g., deaths of soldiers in Starcraft II, revival of enemies in Neural MMO), and although some empty spaces can be set aside in advance from a network compatibility perspective, these spaces are not always suitable for new entities. In addition, the order of the entities should not affect the RL model training. Unlike the order of words in sentences, the order dependence in RL leads to an inability to handle the input of a varying number of entities.

A model for this issue should satisfy two critical requirements. First, the model should be able to process input of any size. The number of input tokens depends on the number of entities. Second, it should be order-agnostic or permutation-invariant. The output of the model should not change under any permutation of the elements in the input. Classical feedforward neural networks violate both requirements, and RNNs are sensitive to input order. To this end, we use the transformer architecture as the core of MAGENTA. We modify the vanilla transformer architecture by removing the positional encoding to satisfy these two requirements.

Furthermore, to force our model to be order agnostic, we design two types of the transformer architecture, as shown in Figure 4: Encoder-Pooling (**EP**) and Encoder-Decoder (**ED**). Detailed despcritption can be found in Appendix. The encoder is used to obtain the mapping between the token representations of entities $\left(e^1, \ldots, e^n\right)$ and the output of the attentive features $\left(\hat{e}^1, \ldots, \hat{e}^n\right)$, where $n$ is the number of all entities. The key idea is that each agent can only affect the related entities, and thus should pay more attention to them.

For **EP**, we use the pooling function to post-process the embeddings. A network that performs pooling over embeddings extracted from the elements of a set. More formally, $\text{net}\left(\left\{\hat{e}^1, \ldots, \hat{e}^n\right\}\right) = \rho\left(\text{pool}\left(\left\{\phi\left(\hat{e}^1\right), \ldots, \phi\left(\hat{e}^n\right)\right\}\right)\right)$. Here, operations like mean, summation, max, or similar can be used as the pooling function to aggregate the embeddings. Zaheer et al. have proven that all permutation-invariant functions can be represented as this inline equation when pool is the sum operator and $\phi, \rho$ any continuous functions, thus satisfying the permutation-invariance requirement.

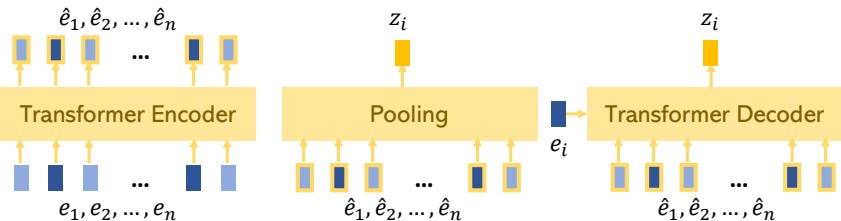

Figure 4: The transformer architecture in MAGENTA. (Left) The transformer encoder. (Mid) Pooling. (Right) The transformer decoder. EP consists of the left and mid parts, while ED consists of the left and right parts. We denote the the token representations of entities by $\left(e^1, \ldots, e^n\right)$ and the output of the attentive features by $\left(\hat{e}^1, \ldots, \hat{e}^n\right)$. We unify $z_i$ from different architectures as the permutation-invariant representation.

For **ED**, we design a decoder to post-process the embeddings. Specifically, instead of using self-attention, the decoder uses the original attention mechanism and outputs $\text{Attention}(Q, K, V) \in \mathbb{R}^{1 \times d_v}$, where $Q$ is the embedding $\hat{e}^i$ of the current agent $i$ (an agent is also an entity), while $K, V$ are the embeddings of all entities. In this way, the output of ED is only one single vector that does not change under any permutation of the elements in the input.

### 4.3 TRAINING AND IMPLEMENTATION DETAILS

**Training scheme.** We focus on MARL scenarios and choose three representative environments from the industrial and academic domains: Honor of Kings (HoK), Neural MMO (NMMO), and Starcraft II micromanagement (SMAC). We design our three-stage training scheme as shown in Fig.5.

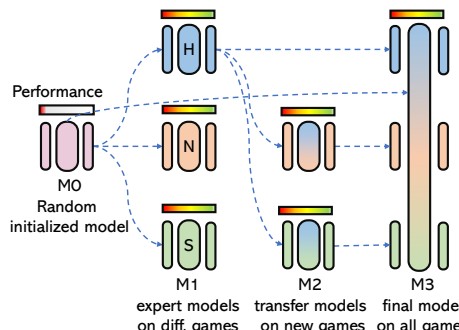

**Stage 1.** From the random initialized model $M^0$, we train expert models with MAGENTA on each individual game and obtain expert models $M_H^1, M_N^1, M_S^1$ in different games, where $H$ represents HoK, $N$ represents NMMO, $S$ represents SMAC.

**Stage 2.** We transfer $M_H^1$ to new games and get $M_{H,N}^2, M_{H,S}^2$. The tokenizer and output are trained from scratch in the new games. Note that HoK, with a state and action space of magnitude $10^{20000}$ (Ye et al., 2020), is much more complex. So we transfer the trained HoK transformer to NMMO and SMAC.

Figure 5: The training scheme of MAGENTA. The dotted lines represent transfers to a new game or a new scenario. Each cell represents tokenizer, transformer and output, respectively. Different colors represent different games.

**Stage 3.** We start from $M^0$ and train the transformer $M^3 = M_{H,N,S}^3$ in selected scenarios in these three environments. The tokenizer and output are from $M^2$. Then we check whether $M^3$ can handle never-seen-before scenarios in these games with few-shot transfer.

**Policy updates.** In order to avoid training instability in our large-scale distributed environment, we use the dual-clip PPO method for each agent. Unlike the original algorithm, we introduce the policy $\pi_\theta (a_i \mid o_i)$ and the estimate of advantages $\hat{A}_t (a_i, o_i)$. When $\pi_\theta (a_i \mid o_i) \gg \pi_{\theta_{\text{old}}} (a_i \mid o_i)$ and $\hat{A}_t < 0$, the ratio $r_t(\theta) = \frac{\pi_\theta(a_i|o_i)}{\pi_{\theta_{\text{old}}}(a_i|o_i)}$ is huge, causing the large and unbounded variance since $r_t(\theta)\hat{A}_t \ll 0$. Dual-clip PPO introduces another clipping parameter $c$ that indicates the lower bound when $\hat{A}_t < 0$. The new objective is defined as:

$$L^{\text{policy}}(\theta) = \hat{\mathbb{E}}_t \left[ \max \left( c\hat{A}_t, \min \left( \text{clip} \left( r_t(\theta), 1 - \tau, 1 + \tau \right) \hat{A}_t, r_t(\theta)\hat{A}_t \right) \right) \right], \tag{1}$$

where $\tau$ is the original clip parameter in PPO.

**Value updates.** We use the multi-head value mechanism for different rewards sources. Specifically, the values of achieving one reward are regarded as one head. Therefore, the value loss is defined as:

$$L^{\text{value}}(\theta) = \hat{E}_t \left[ \sum_{\text{head } k} \left( R_t^k - \hat{V}_t^k \right) \right], V_{\text{total}} = \sum_{\text{head } k} w_k V_t^k (o_i) \tag{2}$$

where $w_k$ is the learnable weight of the $k$-th head and $V_t^k (o_i)$ is the $k$-th value.

## 5 EXPERIMENTS

We conduct our experiments to answer a number of questions:

- How is the performance of the entity transformer in single game?
- How effective is MAGENTA in transfer to **new games**?

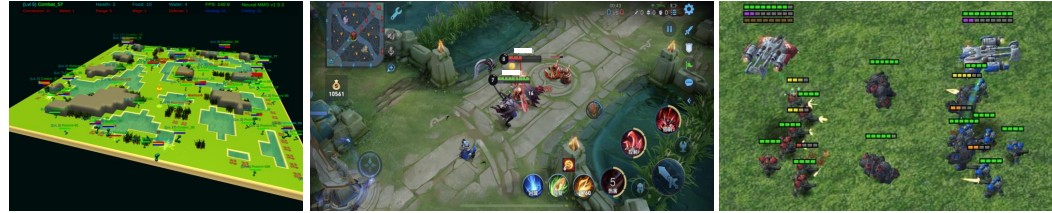

Figure 6: Games where MAGENTA trains. (Left) Neural MMO. (Mid) Honor of Kings. (Right) Starcraft II micromanagement (SMAC). See detailed description about these games in Appendix.

- How effective is MAGENTA in transfer to **new scenarios**?
- How does MAGENTA **scale with model size**?
- What does MAGENTA attend to? (See Appendix E.2 due to page limits.)
- What the learned embedding in games looks like? (See Appendix E.3)

## 5.1 SETUP

**Models used in the experiments.** We list the different versions of /name and the corresponding transfer experiments in Table 2.

| | EP | ED | TL |
|---|---|---|---|
| $M^1$ (baselines) | H,S,N | H,S,N | H |
| $M^2$ (new games) | H-> S,N | H-> S,N | - |
| $M^3$ (new scenarios) | H+S+N-> S,N | H+S+N-> S,N | - |

Table 2: Different Models and Transfer Experiments.

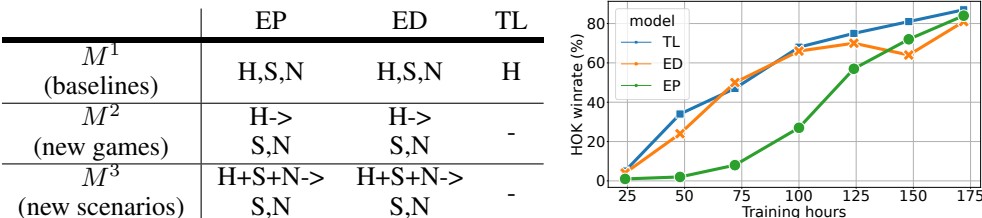

Figure 7: The performance of different variants of MAGENTA in HoK 3v3[2][3].

**Training and fine-tuning.** We train MAGENTA models on Hok with an ADAM optimizer of a $2 \cdot 10^{-4}$ learning rate, $\beta_1 = 0.9$ and $\beta_2 = 0.999$, without weight decay, gradient clip 0.5, and batch size 2048. And we train MAGENTA on SMAC and NMMO with an ADAM optimizer of $1 \cdot 10^{-4}$ learning rate, $\beta_1 = 0.9$ and $\beta_2 = 0.999$, no weight decay, gradient clip 0.3, and batch size 4096. Detailed hyperparameter are described in the Appendix F.

**Metrics.** We measure the performance of MAGENTA in HoK by the win rate and in SMAC/NMMO by cumulative rewards. Due to prohibitively long training times, we only evaluated one training seed in HoK and three training seeds in SMAC/NMMO.

## 5.2 HOW IS THE PERFORMANCE OF THE ENTITY TRANSFORMER?

In this subsection, we mainly evaluate $M^1$ of MAGENTA in HoK 3v3 scenario. We trained 3 different variants of MAGENTA: EP, ED, TL (EP followed by an LSTM network), and train FC+LSTM (FC layers as replacement of the transformer) as the opponent to calculate the win rate with three variants. All variants have similar model sizes. We introduce the TL model to show that MAGENTA is orthogonal to the time-sequential modeling methods. At each point, two models with the same training time compete with each other.

---

[3] We omit the performance of the FC+LSTM model, since it is a 50% line.

[3] We use time-axis since the training is in a distributed manner. It is difficult to show the exact number of samples.

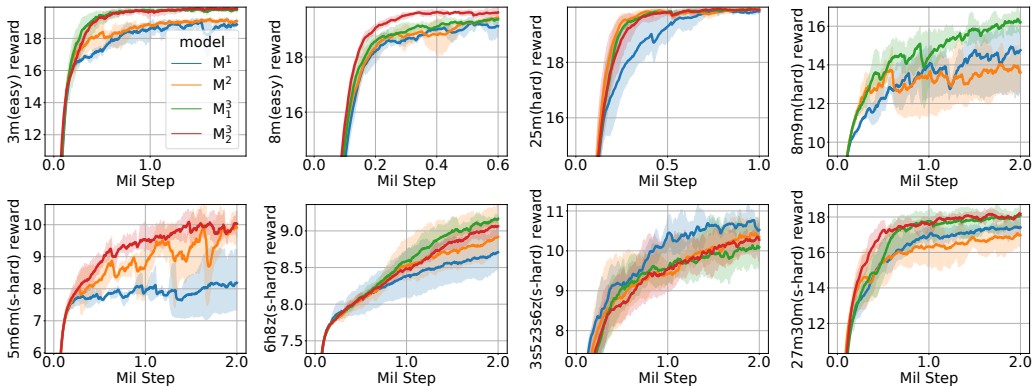

Figure 8: Transfer performance in SMAC. $M^2$ is trained in HoK, $M_1^3$ in HoK, SMAC 5m_vs_6m scenario, NMMO 8 agent scenario, and $M_2^3$ in HoK, SMAC 8m_vs_9m, NMMO 8 agent scenario.

In Fig.7, we can see that after 160 hours of training, all variants of MAGENTA outperform the FC+LSTM model. It shows the benefit of using the transformer architecture. An interesting finding is that, at the beginning, MAGENTA hardly won the FC+LSTM model. The transformer can increase model capacity and expressiveness while needing more training data compared to the FC model.

## 5.3 HOW EFFECTIVE ARE MAGENTA AT TRANSFER TO NEW GAMES?

We want to evaluate whether MAGENTA can adapt the the novel games. To do this, we transfer the trained HoK transformer to NMMO and SMAC. We show the training process of $M^1$, $M^2$ and $M^3$. In Fig.8, we show the result of transfer to SMAC. Before this transfer, $M^2$ is already trained in HoK, $M_1^3$ in HoK, SMAC 5m_vs_6m scenario, NMMO 8 agent scenario, and $M_2^3$ in HoK, SMAC 8m_vs_9m, NMMO 8 agent scenario. For $M^2$, the tokenizer and the output are trained from scratch with the pre-trained transformer.

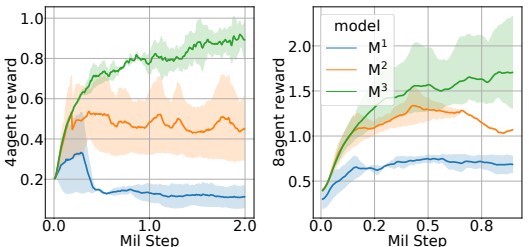

Figure 9: Transfer performance in NMMO.

For $M^3$, all modules in MAGENTA are pre-trained. In this subsection, we focus on $M^1$ and $M^2$, since $M^2$ is trained in the new games. We can see that $M^2$ can outperform $M^1$ in most SMAC scenarios, including easy, hard, and super hard scenarios. In Fig.9, we show the result of transfer to NMMO. We show the performance in the 4 agent scenario and the 8 agent scenario. We can clearly see that $M^2$ can outperform $M^1$. The results verifies our hypothesis that pre-training in other games should indeed help with rapid learning of a new game.

## 5.4 HOW EFFECTIVE IS MAGENTA IN TRANSFER TO NEW SCENARIOS?

We devise our own evaluation setup by transferring to the never-seen-before scenarios in HoK, SMAC, and NMMO. HoK has three scenarios: 1v1, 3v3, and 5v5. In HoK, we have two experiments to answer whether we can transfer to more or fewer agents.

In Fig.10a, we show the training performance of model $M_{HoK}^1$, $M_{1v1}^2$, $M_{3v3}^2$ and $M_{5v5}^2$ [4]. The result shows that the models transferred from 3v3 and 5v5 to 1v1 outperform the model trained in 1v1 from scratch. And there is a slight performance difference between $M_{3v3}^2$ and $M_{5v5}^2$. In Fig.10b, the model transferred from 1v1 to 5v5 outperforms the model trained in 5v5 from scratch. The experiments show that MAGENTA learned some transferable knowledge which can be used in the never-seen-before scenarios, even in the single agent scenario. Then we look back at Fig.8 and Fig.9. We can see that $M^3$ can outperform $M^2$ in the held-out scenarios. The results in this subsection highlight the benefit of MAGENTA's rapid fine-tuning to new scenarios across different tasks and number of agents or entities.

---

[4] We omit other games in the subscript.

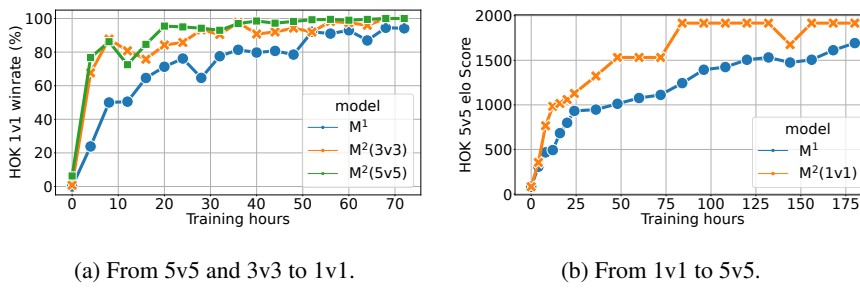

(a) From 5v5 and 3v3 to 1v1.

(b) From 1v1 to 5v5.

Figure 10: The transfer in HoK. The win rate is against the FC+LSTM model.

## 5.5 How does MAGENTA scale with model size?

We obtain the final performance of $M^3$ on all games and choose $M^1$ as the baseline to compute the normalized score. In Fig.11 It shows that when the size of the model increases, in the complex game, i.e., HoK, MA-GENTA has better performance. In less complex games, MAGENTA shows a flat or decreasing performance with increasing model size. We interpret this result as overparameterization, where a richer model is fitting than necessary. In fact, we conducted a confirmatory experiment using just a few FC layers in SMAC and NMMO. It can show a performance similar to $M^1$.

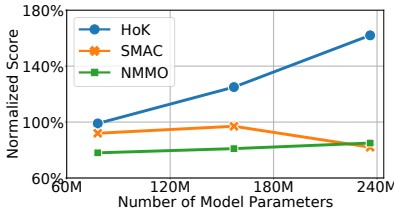

Figure 11: Normalized scores for all training games with different model sizes.

We also conduct an ablation study in HoK on which parts of MAGENTA are transferable in Fig.12. We transfer different parts of MAGENTA to the new scenarios. It shows that all transfers benefit the adapt to the new scenarios and the transfer of the whole model weights achieves the best performance.

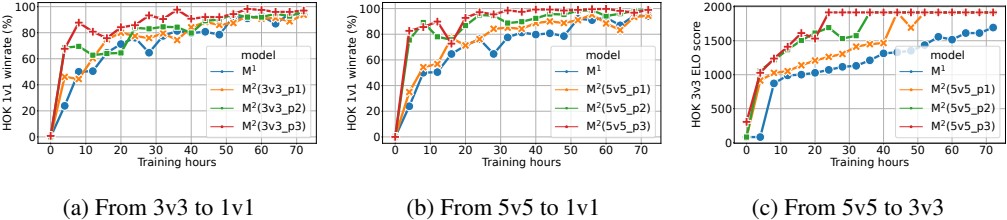

(a) From 3v3 to 1v1

(b) From 5v5 to 1v1

(c) From 5v5 to 3v3

Figure 12: The ablation of transferable parts of MAGENTA in HoK. P1 represents the transfer of tokenizer. P2 represents the transfer of tokenizer and transfomer. P3 represents the transfer of the whole model weights.

## 6 Conclusion

In this paper, we have made an attempt to develop highly capable generalist agents. Namely, we provide a perspective: viewing different games as different languages. When different tokenizers are trained for various games, entities are split into tokens. The utilization of transformer models can deal with different feature inputs and various number of entities. The attention mechanism in the transformer also encourages the agent to focus on highly related entities. Our results exhibit a clear benefit of using large transformer-based models from entity perspective in multi-agent multi-game domains. We believe the trends suggest clear paths for future work, that with larger models and larger suites of tasks.

**Limitations.** We acknowledge some limitations of our conclusions. First, our results are largely based on performance in battle games, such as real-time strategy games, MOBA games, and MMO games, where the rules are similar to each other. Second, the tokenizer is not general for all games. We still need to train specific tokenizers to handle different entities in various games. Third, we still follow the online fashion, which remains the question whether offline datasets can help and boost the training efficiency. Lastly, it remains unclear whether we can observe other forms of generalization, such as zero-shot adaptation, as well as whether our conclusions hold for other settings.

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

## A    RELATED WORK ABOUT RL IN VIDEO GAMES

Games have been a focus of artificial intelligence research for decades as a stepping stone towards more general applications. Recently, (deep) RL has been widely used in Game AI, for example, Go (Silver et al., 2016), Atari (Mnih et al., 2015), Super Mario Bros (Pathak et al., 2017), Quake III Arena Capture the Flag (Jaderberg et al., 2019), VizDoom (Kempka et al., 2016), Minecraft (Research, 2019; Guss et al., 2019), Neural MMO (Suarez et al., 2021), Honor of Kings (Wei et al., 2022; Ye et al., 2020; Gao et al., 2021), StarCraft II (Jaderberg et al., 2019; Samvelyan et al., 2019; Vinyals et al., 2017), and Dota 2 (OpenAI, 2019). In this work, we mainly focus on using a single set of weights to play multi-scenario multi-agent games, i.e. HoK, Neural MMO, and StarCraft II.

## B    HONER OF KINGS

**Honor of Kings** is one of the most popular MOBA games worldwide. The gameplay is to divide ten players into two camps to compete on the same symmetrical map. Players of each camp compete for resources through online confrontation, team collaboration, etc., and finally win the game by destroying the enemy's crystal. The behaviors performed by players in the game can be divided into two categories: macro-strategies and micro-operations. **Macro-strategy** that is, long-distance scheduling or collaborating with teammates for quick resource competition, such as long-distance support for teammates, collaborating to compete for monster resources, etc. **Micro-operation** that is, the real-time behavior adopted by each player in various scenarios, such as skill combo release, evading enemy skills, etc. Complicated game maps, diverse hero combinations, diverse equipment combinations, and diverse player tactics make MOBA games extremely complex and exploratory.

Figure 13 shows the UI interface of *Honor of Kings*. For fair comparisons, all experiments in this paper were carried out using a fixed released gamecore version (Version 3.73 series) of *Honor of Kings*.

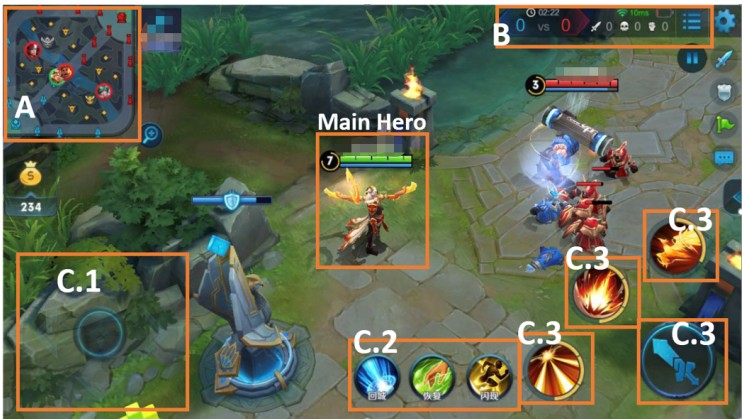

Figure 13: **The UI interface of *Honor of Kings*.** The hero controlled by the player is called *Main Hero*. The player controls the hero's movement through the bottom-left wheel (C.1) and releases the hero's skills through the bottom-right buttons (C.2, C.3). The player can observe the local view via the screen, observe the global view via the top-left mini-map (A), and obtain game states via the top-right dashboard (B).

**Feature Design.** See Table 3.

**Agent Action.** Table 4 shows the action space of agents.

**Reward Design.** Table 5 demonstrates the details of the designed environment reward.

## C    SMAC

SMAC benchmark is a challenging set of cooperative StarCraft II maps for micromanagement developed by (Samvelyan et al., 2019) built on DeepMind's PySC2 (Vinyals et al., 2017). We

Table 3: Feature details.

| Feature Class | Field | Description | Dimension |
|---|---|---|---|
| **1. Unit feature** | Scalar | Includes heroes, minions, monsters, and turrets | 3946 |
| Heroes | Status | Current HP, mana, speed, level, gold, KDA, and magical attack and defense, etc. | 1562 |
| | Position | Current 2D coordinates | 20 |
| Minions | Status | Current HP, speed, visibility, killing income, etc. | 920 |
| | Position | Current 2D coordinates | 80 |
| Monsters | Status | Current HP, speed, visibility, killing income, etc. | 728 |
| | Position | Current 2D coordinates | 56 |
| Turrets | Status | Current HP, locked targets, attack speed, etc. | 540 |
| | Position | Current 2D coordinates | 40 |
| **2. In-game stats feature** | Scalar | Real-time statistics of the game | 104 |
| Static statistics | Time | Current game time | 57 |
| | Camp | Types of two camps | 1 |
| | Alive heroes | Number of alive heroes of two camps | 10 |
| | Kill | Kill number of each camp | 6 |
| | Alive turrets | Number of alive turrets of two camps | 8 |
| Comparative statistics | Alive heroes diff | Alive heroes difference between two camps | 11 |
| | Kill diff | Kill difference between two camps | 5 |
| | Alive turrets diff | Alive turrets difference between two camps | 6 |

Table 4: The action space of agents.

| Action | Detail | Description |
|---|---|---|
| What | Illegal action | Placeholder. |
| | None action | Executing nothing or stopping continuous action. |
| | Move | Moving to a certain direction determined by move x and move y. |
| | Normal Attack | Executing normal attack to an enemy unit. |
| | Skill1 | Executing the first skill. |
| | Skill2 | Executing the second skill. |
| | Skill3 | Executing the third skill. |
| | Skill4 | Executing the fourth skill (only a few heroes have Skill4). |
| | Summoner ability | An additional skill choosing before the game begins (10 to choose). |
| | Return home(Recall) | Returning to spring, should be continuously executed. |
| | Item skill | Some items can enable an additional skill to player's hero. |
| | Restore | Blood recovering continuously in 10s, can be disturbed. |
| | Collaborative skill | Skill given by special ally heroes. |
| How | Move X | The x-axis offset of moving direction. |
| | Move Y | The y-axis offset of moving direction. |
| | Skill X | The x-axis offset of a skill. |
| | Skill Y | The y-axis offset of a skill. |
| Who | Target unit | The game unit(s) chosen to attack. |

introduce **states and observations**, **action space** and **rewards** of SMAC, and **environmental settings of MAGENTA** below.

**States and Observations.** At each time step, agents receive local observations within their field of view, which contains information (distance, relative x, relative y, health, shield, and unit type) about the map within a circular area for both allied and enemy units and makes the environment partially observable for each agent. The global state is composed of the joint observations, which could be used during training. All features, both in the global state and in individual observations of agents, are normalized by their maximum values.

**Action Space.** actions are in the discrete space. Agents are allowed to make move[direction], attack[enemy id], stop and no-op. The no-op action is only for dead agents and it is the only legal action for them. Agents can only move in four directions: north, south, east, or west. The shooting range is set to for all agents. Having a larger sight range than a shooting range allows agents to make use of the move commands before starting to fire. The automatical built-in behavior of agents is also disabled for training.

Table 5: The details of the environment reward.

| Head | Reward Item | Weight | Type | Description |
|------|-------------|--------|------|-------------|
| Farming Related | Gold | 0.005 | Dense | The gold gained. |
| | Experience | 0.001 | Dense | The experience gained. |
| | Mana | 0.05 | Dense | The rate of mana (to the fourth power). |
| | No-op | -0.00001 | Dense | Stop and do nothing. |
| | Attack monster | 0.1 | Sparse | Attack monster. |
| KDA Related | Kill | 1 | Sparse | Kill a enemy hero. |
| | Death | -1 | Sparse | Being killed. |
| | Assist | 1 | Sparse | Assists. |
| | Tyrant buff | 1 | Sparse | Get buff of killing tyrant, dark tyrant, storm tyrant. |
| | Overlord buff | 1.5 | Sparse | Get buff of killing the overlord. |
| | Expose invisible enemy | 0.3 | Sparse | Get visions of enemy heroes. |
| | Last hit | 0.2 | Sparse | Last hitting an enemy minion. |
| Damage Related | Health point | 3 | Dense | The health point of the hero (to the fourth power). |
| | Hurt to hero | 0.3 | Sparse | Attack enemy heroes. |
| Pushing Related | Attack turrets | 1 | Sparse | Attack turrets. |
| | Attack crystal | 1 | Sparse | Attack enemy home base. |
| Win/Lose Related | Destroy home base | 2.5 | Sparse | Destroy enemy home base. |

**Rewards.** At each time step, the agents receive a joint reward equal to the total damage dealt on the enemy agents. In addition, agents receive a bonus of 10 points after killing each opponent, and 200 points after killing all opponents for winning the battle. The rewards are scaled so that the maximum cumulative reward achievable in each scenario is around 20.

**Environmental Settings of MAGENTA.** The difficulty level of the built-in game AI we use in our experiments is level 7 (very difficult) by default as many previous works did. The used scenarios are shown in Table 6. We present the table of all scenarios in SMAC in Table 6. The *Ally Units* are agents trained by MARL methods and *Enemy Units* are built-in game bots. For example, 5m_vs_6m indicates that the number of MARL agent is 5 while the number of the opponent is 6. The agent (unit) type is *marine*[5]. This asymmetric setting is hard for MARL methods.

Table 6: SMAC Environments

| Name | Ally Units | Enemy Units | Type |
|------|-----------|-------------|------|
| 3m | 3 Marines | 3 Marines | homogeneous & symmetric |
| 8m | 8 Marines | 8 Marines | homogeneous & symmetric |
| 25m | 25 Marines | 25 Marines | homogeneous & symmetric |
| 5m_vs_6m | 5 Marines | 6 Marines | homogeneous & asymmetric |
| 8m_vs_9m | 8 Marines | 9 Marines | homogeneous & asymmetric |
| 27m_vs_30m | 27 Marines | 30 Marines | homogeneous & asymmetric |
| 3s5z_vs_3s6z | 3 Stalkers & 5 Zealots | 3 Stalkers & 6 Zealots | heterogeneous & asymmetric |
| 6h_vs_8z | 6 Hydralisks | 8 Zealots | micro-trick: focus fire |
| corridor | 6 Zealots | 24 Zerglings | micro-trick: wall off |

## D   NEURAL MMO

Neural MMO is a platform inspired by Massively Multiplayer Online games, a genre that simulates persistent worlds with large player populations and diverse gameplay objectives. It features game systems configurable to research both on individual aspects of intelligence (e.g. navigation, robustness, collaboration) and on combinations thereof. Support spans 1 to 1024 agents and minute- to hours-long time horizons. Our enviroment code is heavily based on

[5]A type of unit (agent) in StarCraft II. Readers can refer to https://liquipedia.net/starcraft2/Marine_(Legacy_of_the_Void) for more information

the IJCAI2022-Neural MMO PvE challenge: https://www.aicrowd.com/challenges/ijcai-2022-the-neural-mmo-challenge.

**Observations and Actions:** Neural MMO agents observe sets of *objects* parameterized by discrete and continuous *attributes* and submit lists of *actions* parameterized by lists of discrete and object-valued *arguments*. This parameterization is flexible enough to avoid major constraints on environment development and amenable to efficient serialization (see documentation) to avoid bottlenecking simulation. Each observation includes 1) a fixed crop of *tile* objects around the given agent parameterized by *position* and *material* and 2) the other *agents* occupying those tiles parameterized by around a dozen properties including current *health*, *food*, *water*, and *position*. Agents submit *move* and *attack* actions on each timestep. The *move* action takes a single *direction* argument with fixed values of *north*, *south*, *east*, and *west*. The *attack* action takes two arguments: *style* and *target*. The *style* argument has fixed values of *melee*, *range*, and *mage*. The agents in the current observation are valid *target* argument values. Encoding/decoding layers are required to project the hierarchical observation space to a fixed length vector and the flat network hidden state to multiple actions. We also provide reusable PyTorch subnetworks for these tasks.

Neural MMO tasks are defined by a reward function on a particular environment configuration (as per above). Users may create their own reward functions with full access to game state, including the ability to define per-agent reward functions. We also provide two default options: a simple survival reward (-1 for dying, 0 otherwise) and a more detailed achievement system. Users may select between *self-contained* and *tournament* evaluation modes, depending on their research agenda.

**Achievement system:** This reward function is based on gameplay milestones. For example, agents may receive a small reward for obtaining their first piece of armor, a medium reward for defeating three other players, and a large reward for traversing the entire map. The tasks and point values themselves are clearly domain-specific, but we believe this achievement system has several advantages compared to traditional reward shaping. First, agents cannot farm reward – in contrast to traditional reward signals, each task may be achieved only once per episode. Second, this property should make the achievement system less sensitive to the exact point tuning. Finally, attaining a high achievement score somewhat guarantees complex behavior since tasks are chosen based upon difficulty of completion. We are currently running a public challenge that requires users to optimize this metric.

# E   ADDITIONAL EXPERIMENTAL RESULTS

## E.1   RAW SCORES OF $M_3$ IN HOK, SMAC, AND NMMO

We show the raw scores of $M_3$ in HoK, SMAC, and NMMO. In SMAC, we compute the mean reward over 8 environments and NMMO over 2 environments.

|      | HoK (win rate) | SMAC (mean reward over 8 envs) | NMMO (mean reward over 2 envs) |
|------|----------------|--------------------------------|--------------------------------|
| 78M  | 69.3%          | 15.6                           | 1.14                           |
| 157M | 87.5%          | 16.4                           | 1.19                           |
| 236M | 97.2%          | 13.9                           | 1.25                           |

## E.2   WHAT DOES MAGENTA ATTEND TO?

We find that MAGENTA consistently attends to related entities. Fig.14 visualizes the selected attention heads and layers for HoK 3v3 scenarios. By fixing one frame in a running competition in HoK, we find that heads attend to entities such as monster, enemy, and friend, which are the entities highly related to the current agent's state. The visualization of MAGENTA in SMAC is shown in Appendix 15. We find that heads attend to allies.

## E.3   WHAT THE LEARNED EMBEDDING IN GAMES LOOK LIKE?

In this subsection, we visualize the embedded entities in HoK and SMAC using t-SNE (Van der Maaten & Hinton, 2008). We run MAGENTA for one episode in HoK and collect the embedding of entities for 6000 game frames. Also, we run MAGENTA for 50 episodes in SMAC and collect

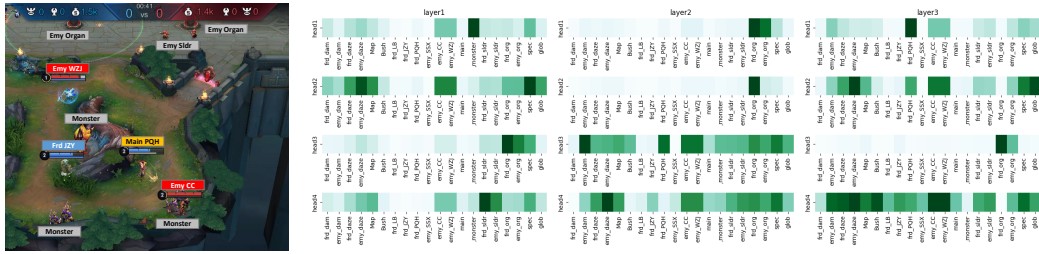

Figure 14: The attention of the entity transformer on 3v3

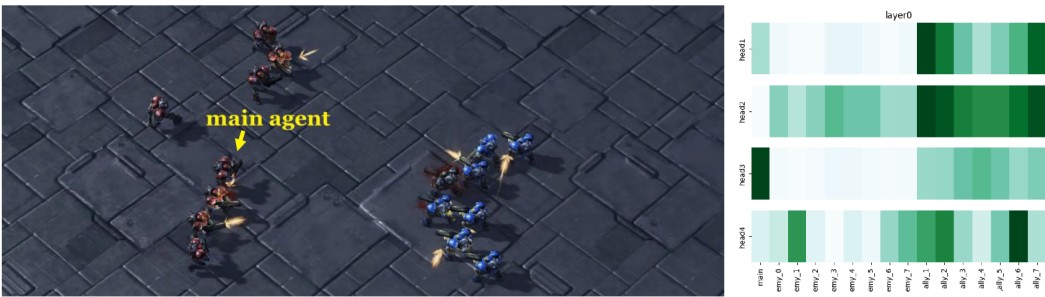

Figure 15: The visualization of attention weights in SMAC.

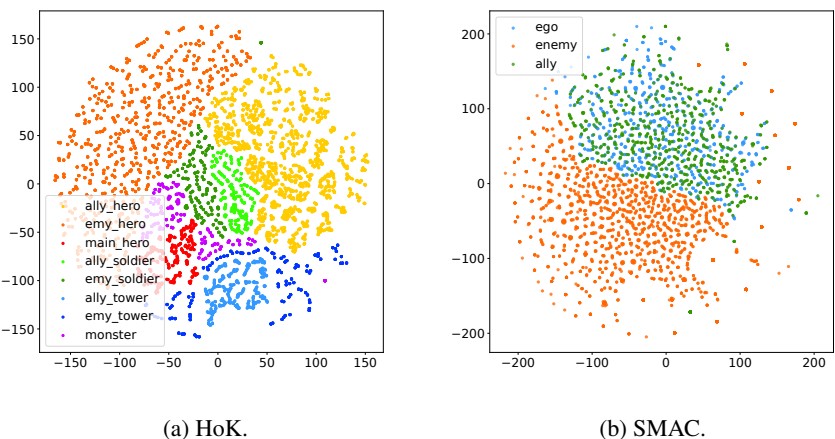

(a) HoK.

(b) SMAC.

Figure 16: The visualization of entity embedding in HoK and SMAC.

the embedding of the entities. We can see a clear separation between allies' embedding and enemies' embedding in both HoK and SMAC. Even though HoK and SMAC are two different games, MAGENTA can still learn some general knowledge across different games.

## F  IMPLEMENTATION DETAILS

First, we introduce the EP and EP architecture. The parameters of the network architecture are: Transformer Encoder layers: 3, Decoder layers: 3, HeadNum: 4, HeadDim: 256. $z$ is a vector with a dimension of $d_v$. $d_v$ is the dimension of the value matrix in the attention module. Since the output of the encoder is $(n, d_v)$, where n is the number of entities, the pooling merges along the n entities and the decoder uses $e_i$ as a query to extract information. Here we use $z_i$ to denote the

permutation-invariant representation and do not differentiate $z_i$ from different models. We modify the main text according to this point for clarity.

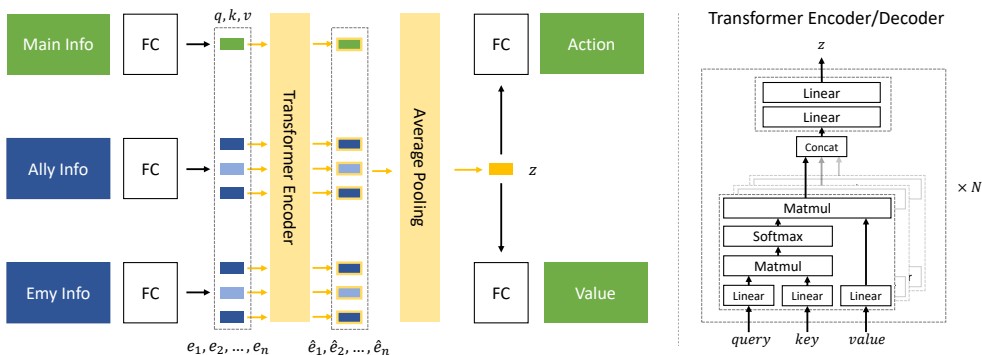

Figure 17: The architecture of EP.

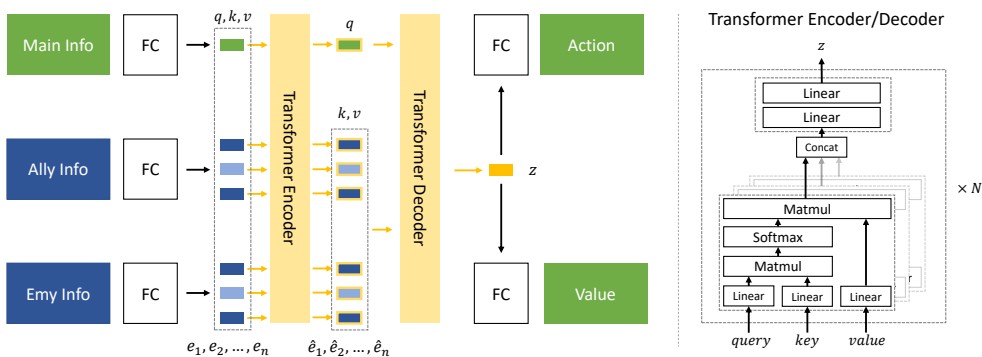

Figure 18: The architecture of ED.

Here, we describe the framework. For HoK, we use the self-developed RL framework for training (Ye et al., 2020; Gao et al., 2021). For SMAC and NMMO, we use the open-source Ray RLlib implementation of Proximal Policy Optimization (PPO), which scales out using multiple workers for experience collection. This allows us to use a large amount of rollouts from parallel workers during training to ameliorate high variance and aid exploration. We do multiple rollouts in parallel with distributed workers and use parameter sharing for each agent. The trainer broadcasts new weights to the workers after their synchronous sampling. Now we turn our attention to environment-specific settings.

Table 7: MAGENTA hyper-parameters used in HoK.

| Name | Value |
| --- | --- |
| Discount rate | 0.99 |
| GAE parameter | 1.0 |
| KL coefficient | 0.2 |
| Rollout fragment length | 1000 |
| Training batch size | 100000 |
| SGD minibatch size | 10000 |
| # of SGD iterations | 60 |
| Learning rate | 1e-4 |
| Entropy coefficient | 0.0 |
| Clip parameter | 0.3 |
| Value function clip parameter | 10.0 |

Table 8: MAGENTA hyper-parameters used in SMAC.

| Name | Value |
|------|-------|
| Discount rate | 0.99 |
| GAE parameter | 1.0 |
| KL coefficient | 0.2 |
| Rollout fragment length | 1000 |
| Training batch size | 100000 |
| SGD minibatch size | 10000 |
| # of SGD iterations | 60 |
| Learning rate | 1e-4 |
| Entropy coefficient | 0.0 |
| Clip parameter | 0.3 |
| Value function clip parameter | 10.0 |

Table 9: MAGENTA hyper-parameters used in NMMO.

| Name | Value |
|------|-------|
| Discount rate | 0.99 |
| GAE parameter | 1.0 |
| KL coefficient | 0.2 |
| Rollout fragment length | 1000 |
| Training batch size | 100000 |
| SGD minibatch size | 10000 |
| # of SGD iterations | 60 |
| Learning rate | 1e-4 |
| Entropy coefficient | 0.0 |
| Clip parameter | 0.3 |
| Value function clip parameter | 10.0 |

