# OpenReview forum: "Multi-Agent Multi-Game Entity Transformer"
_ICLR.cc/2023/Conference — Submitted to ICLR 2023_

### Official Review · Reviewer_gmYJ · 2022-10-19

**Confidence:** 3
**Correctness:** 3
**Technical Novelty And Significance:** 3
**Empirical Novelty And Significance:** 2
**Recommendation:** 6

**Clarity, Quality, Novelty And Reproducibility:**

My primary concerns with this paper are related to clarity, and sometimes quality (in the sense of correctness).

## Detailed comments

- "we analogize games as languages" (in abstract) --> I think this is potentially confusing. When I first read this, I thought it would mean that the authors would use a single language in which to describe (the rules of) the different games. But the only relation to languages seems to be that tokenization is also commonly-used there... which hardly seems an important enough similarity to describe it as "analogize" like this.
- "due to trail-and-error, RL is inclined to overfit to training environments" --> how is this related to trial-and-error? If, hypothetically, we could exhaustively enumerate the complete space of all possible trajectories and learn from that (so no more need for trial-and-error), would we not still overfit to the environment for which we trained anymore?
- "we investigate whether a single model, with a single set of parameters, can be trained by playing multiple multi-agent games in an online manner." --> this sentence highly suggests, especially due to the phrasing and mention of a single set of parameters, that a single model will be trained that can play *all* games at the same time, with the same weights (i.e. not separated by training runs in between the different games). If I understood all the experiments correctly, there is no experiment where truly the exact same set of weights is used for multiple different games. I have a similar issue with describing the agents as "generalist agents" in the Conclusion. This is an important difference with agents such as Gato.
- Throughout most, or maybe all, of the paper, there are several mentions of permutation invariance, but it is not actually clear in what ways there is permutation invariance: for which dimensions of the input space do we want permutation invariance, and why? After having read the full paper I can understand that the permutation invariance is between different entities: we should not care about the order in which different entities are presented as inputs to the model. But this is not clearly described in the paper, at least not nearly early enough. This intuition is very simple, and would be very helpful to describe early in the paper, but it isn't. In contrast, the phrase "permutation-invariant" already does show up several times early in the paper (e.g., towards the end of the Introduction), but without the intuition about the dimensions of the input space for which we want permutation invariance, and why, this is difficult to follow.
- It is not clear to me how "words" are "typically composed of images and vectors". It would probably help to give some examples of what things would be images, and what things would be vectors.
- One subfigure of Figure 8 is missing the red line, and one is missing the green line. Where did they go?

## Other minor comments

- "(2020), Furthermore" --> looks like a comma instead of a period ending the sentence
- "AlphaStar requires [...] half a month" --> should provide references for all these agents and numbers
- many instances of "casual" which I think should probably be "causal"?
- several cases of "radio" which I think should be "ratio"?
- Subsection 3.3 just lists a bunch of publications that used video games, but does not discuss that work in any way. In this form, I'd say the entire subsection seems unnecessary and could be removed.
- "can be represented as (1)" --> what is (1)?
- Legends in Figures 8 and 9 are impossible to read unless I zoom in to 150%, which probably means they would be unreadable in print.
- The caption of Figure 12 mentions P3, but I dont see P3 anywhere.

**Strength And Weaknesses:**

## Strengths:

The primary strength of the paper is that the empirical results look convincing, and sufficiently extensive with some ablations. To the best of my knowledge, the proposed architecture is also novel, and it has some interesting insights.

## Weaknesses:

The primary weakness is that several parts of the paper are unclear, and some statements even seem wrong. See below for detailed comments related to this.

---

**Note after discussion with authors**: I feel that the paper has been substantially improved during the discussion phase and have updated my score accordingly.

**Summary Of The Paper:**

This paper describes a tranformer-based architecture for policies for multi-agent (video) games, that tokenises image-based inputs plus entity-component-system-based inputs (representing data of entities in video games), allowing the architecture to process inputs from various different games in a single and consistent manner. Policies are trained in three different video games (Honor of Kings, Neural MMO, and StarCraft II Micromanagement), some of which have multiple different scenarios. Several experiments show that it is effective to train in some games, transfer to others, and then finetune in the other games.

**Summary Of The Review:**

The paper is interesting and has some interesting and potentially significant contributions, but in its current form has too many issues in terms of clarity.

**Note after discussion with authors**: I feel that the paper has been substantially improved during the discussion phase and have updated my score accordingly.

---

> ### Author Response · Authors · 2022-11-14
> **Response to Reviewer gmYJ**
>
> Thank you for carefully reviewing our paper! We greatly appreciate your feedback. Please see below our responses to your comments.
>
> ---------------
>
> **Q1**: Analogize games as languages
>
> **A1**: We align one single game to one single language as shown in Fig.3. Different languages have different tokenizers, so do games. In this case, our transformer can be viewed as a multilingual transformer to capture the common knowledge among different games.
>
> ---------------
>
> **Q2**: Trial-and-error
>
> **A2**: Here we mention trial-and-error since the training environment and test environment are the same in the online RL training. The interaction and feedback are from the same environments. In this case, The trained RL model overfits the training environment while lacking generality to other environments. If we could exhaustively enumerate the complete space of all possible trajectories of those environments, we would still overfit the environment since the RL model does not see other environments.
>
> ----------------
>
> **Q3**: The exact same set of weights is used for multiple different games
>
> **A3**: We want to clarify that actually we only share the parameters of the transformer of MAGENTA among different games. In Sec 5.5, we show a steady performance of M3 with 1,2,3 layer of transformer encoder, respectively. Since the transformer of M3 is with the same weights, Fig. 11 shows that the same set of weights of the transformer is used for multiple different games.
>
> ----------------
>
> **Q4**: Permutation invariance
>
> **A4**: The permutation invariance happens along the entity dimension since different games have different entities and we cannot expect a fixed order of entities in different games. In one single frame of the game, the order of entities' features can be arbitrary. We modify the introduction and Sec 4.2 for more clarity.
>
> ----------------
>
> **Q5**: Some examples of what things would be images, and what things would be vectors.
>
> **A5**: Images: when playing games, we can have a screenshot as an image. Typically we call them frames as shown in [1]. Another example of images is the obstacles in the vision and rendering images of the hero's skills. Vectors: the numerical attributes of the entities can be vectors. For example, a vector composed of the health point, the mana point, the speed, the attack, and defense, etc. The detailed feature can be found in Appendix.
>
> ---------------
>
> **Q6**: One subfigure of Figure 8 is missing the red line, and one is missing the green line.
>
> **A6**: Fig.8 shows the transfer performance. Since $M^3_1$ is trained in HoK, SMAC **5m_vs_6m**, NMMO 8 agents, the green line is not included in the plot of the **5m_vs_6m** scenario. So does the red line.
>
> ---------------
>
> **Q7**: Comments about figures legends and captions.
>
> **A7**: We fix these issues and typos in the revised version.

---

> > ### Comment · Reviewer_gmYJ · 2022-11-15
> > **Thanks for your response**
> >
> > While some of your clarifications and updates to the paper definitely help, a couple of issues still remain.
> >
> > > **Q2**: Trial-and-error
> > >
> > > **A2**: Here we mention trial-and-error since the training environment and test environment are the same in the online RL training. The interaction and feedback are from the same environments. In this case, The trained RL model overfits the training environment while lacking generality to other environments. If we could exhaustively enumerate the complete space of all possible trajectories of those environments, we would still overfit the environment since the RL model does not see other environments.
> >
> > "Trial-and-error" does not mean the same as "training environment and test environment are the same". I could do trial-and-error training in one environment, and then test in another environment. Trial-and-error just means that we learn by doing (by selecting actions and observing what happens). It is incorrect to say that overfitting is related to trial-and-error training.
> >
> > > **Q5**: Some examples of what things would be images, and what things would be vectors.
> > >
> > > **A5**: Images: when playing games, we can have a screenshot as an image. Typically we call them frames as shown in [1]. Another example of images is the obstacles in the vision and rendering images of the hero's skills. Vectors: the numerical attributes of the entities can be vectors. For example, a vector composed of the health point, the mana point, the speed, the attack, and defense, etc. The detailed feature can be found in Appendix.
> >
> > I had already guessed it would be something like this. But this needs to be clarified and explained in the paper, such that future readers of the paper can easily understand it and won't have to guess, rather than just explained in a response to me on OpenReview.
> >
> > > Q7: Comments about figures legends and captions.
> > >
> > > A7: We fix these issues and typos in the revised version.
> >
> > There is still at least one case of "radio" instead of "ratio" in the current revision.

---

> > > ### Author Response · Authors · 2022-11-16
> > > **Re: Thanks for your response**
> > >
> > > Dear Reviewer gmYJ,
> > >
> > > We appreciate your quick response to help us to improve this work. And thanks for pointing out the issue about trial-and-error and the clarity issue.
> > >
> > > Based on your suggestion, we:
> > >
> > > - Modify the statement in the Introductioon by saying: First, when the training environment and the test environment are the same, RL is inclined to overfit the training environments, while lacking generalizability to unknown environments.
> > >
> > > - Add the examples of images and vectors in the Sec 4.1 and Appendix.
> > >
> > > - We again check our typos and fix them.
> > >
> > > We thank you again and if you have any further questions or comments, please post them and we will be happy to have further discussions.
> > >
> > > Paper2668 authors

---

### Official Review · Reviewer_D4zs · 2022-10-23

**Confidence:** 4
**Correctness:** 4
**Technical Novelty And Significance:** 2
**Empirical Novelty And Significance:** 2
**Recommendation:** 5

**Clarity, Quality, Novelty And Reproducibility:**

The writing is mostly clear with some details missing. The results and ablation studies are plenty. The originality of the work seems minor since the authors test only minor modifications to the transformer architecture.

**Strength And Weaknesses:**

The strengthnesses of the paper:
1) Proposed a single transformer architecture to solve multiple different battle games.
2) Ablation studies to test the performance of the model(s) at different training stages.

The weaknesses of the paper:
1) The clarity of the paper can be improved. For example in section 4.2 authors discuss the encoder-pooling and the modified encoder-decoder architectures, but leave many missing details: the latent variable z_i seems to be a scalar with encoder pooling due to the symmetry requirements, which looks rather stranger (or is that the case?).
2) Need more details on the architecture, i.e. how many encoders/decoders/attention heads are used, etc.
3) Also, the performance comparison plots use different x-values: some use samples and some use time. This is a bit confusing.


**Summary Of The Paper:**

In this paper, the authors proposed a novel transformer architecture to solve multi-agent games. The model contains two major parts (1) game specific tokenizers and output layers which encode different game observations into the same token space and output the right actions (2) the main transformer model with ​​permutation invariant pooling and encoder-decoder modifications. With a multi-stages training, the proposed method is able to fast adapt to novel games or new game settings.


**Summary Of The Review:**

While the paper is mostly clearly written, the novelty of the paper is limited.

---

> ### Author Response · Authors · 2022-11-14
> **Response to Reviewer D4zs**
>
> Thank you for carefully reviewing our paper! We greatly appreciate your feedback. Please see below our responses to your comments.
>
> ---------------
>
> **Q1**: Details about the encoder-pooling and the modified encoder-decoder architectures
>
> **A1**: $z$ is a vector with a dimension of $d_v$. $d_v$ is the dimension of the value matrix in the attention module. Since the output of the encoder is $(n, d_v)$, where n is the number of entities, the pooling merges along the n entities and the decoder uses $e_i$ as a query to extract information. Here we use $z_i$ to denote the permutation-invariant representation and do not differentiate $z_i$ from different models. We modify the main text according to this point for clarity.
>
> ---------------
>
> **Q2**: More details on the architecture
>
> **A2**: We include the details on the architecture in the Appendix F.
>
> ----------------
>
> **Q3**: The performance comparison plots use different x-values: some use samples and some use time. This is a bit confusing.
>
> **A3**: Plots in HoK use the time-axis since we use thousands of CPU cores and hundreds of A100 GPUs and the throughput is about 100k samples per min with a distributed training system. It is hard to show the exact number of samples. Plots in SMAC/NMMO use sample-axis since the experiments are relatively small-scale and we conduct them with RLlib for reproducibility.
>
> ----------
>
> **Q4**: Novelty
>
> **A4**: Our aim is to build generalist agents for multiple multi-agent environments. Even though the methodological backbone of MAGENTA is PPO+transformer, which is not novel in the RL community. We provide a new perspective of analogizing games as languages and propose novel architectures for permutation-invariance.

---

> ### Author Response · Authors · 2022-11-25
> **Follow-up Response**
>
> Dear Reviewer D4zs,
>
> We thank you again for recognizing the contribution of our work to the RL/MARL community. We appreciate your time and hard work in providing valuable comments for our paper. Please let us know if you have any questions. More questions and discussions on our work are always welcome!
>
> Sincerely yours,
>
> Paper 2668 authors

---

### Official Review · Reviewer_5qFD · 2022-10-25

**Confidence:** 3
**Correctness:** 4
**Technical Novelty And Significance:** 3
**Empirical Novelty And Significance:** 3
**Recommendation:** 6

**Clarity, Quality, Novelty And Reproducibility:**

I thought it was quite clear. The authors do not have code and do experiments on large games, so I don't think it is very reproducible.

**Strength And Weaknesses:**

I think this is a nice paper. Although similar research has been done in language, offline RL, and multi-game RL, this is the first to show that the same trend of scale helping generalization also works in cooperative multiagent RL.
The experiments are on very large video games, and it is not obvious that so much would transfer to different video games, especially because the representation is not the part that gets transferred, but the transformer that decides which actions to take.
One thing I would have liked to seen would be better small-scale experiments where comparisons with many baselines could be tested.
The results are not striking, but I think they clearly show the benefit of MAGENTA
I think this paper will have some impact in the related literature on cooperative RL and transformers for decision making
Figure 7 should be clearer, I think this is head to head against the FC+LSTM


**Summary Of The Paper:**

This paper introduces MAGENTA, which is a variant of PPO with a transformer architecture. Based on experiments in large cooperative video games, MAGENTA appears to be able to transfer knowledge across games.


**Summary Of The Review:**

I think this paper is the first to show what many might have suspected: that large models transfer and generalize in cooperative multi-agent environments. This is interesting and I think future work can build off of this fact.

---

> ### Author Response · Authors · 2022-11-14
> **Response to Reviewer 5qFD**
>
> Thank you for carefully reviewing our paper! We greatly appreciate your feedback. Please see below our responses to your comments.
>
> ----------
>
> **Q1**:  It is not obvious that so much would transfer to different video games, especially because the representation is not the part that gets transferred, but the transformer that decides which actions to take.
>
> **A1**: The transformer mainly learns the permutation-invariant representations that can be transferred among different games. The output module uses such representations to decide which actions to take. The parameters of the transformer are shared in different games, while the parameters of different tokenizers and output modules are not shared.
>
> ----------
>
> **Q2**: Small-scale experiments
>
> **A2**: Experiments in SMAC and NMMO are small-scale environments related to HoK. MAGENTA is a transformer-based model and can be combined with other MARL algorithms other than PPO. We leave the transferability among different algorithms as future work.
>
> ----------
>
> **Q3**: Figure 7
>
> **A3**: We mentioned this point in Sec 5.2 and footnote 2. We modify the title of Fig.7 in the revised version and the footnote to clarify this point.
>
> ----------
>
> **Q4**: Reproducibility
>
> **A4**: We uploaded our source code in the supplementary materials. We will opensource our code and model parameters after the double-blind review process.

---

> ### Author Response · Authors · 2022-11-25
> **Follow-up Response**
>
> Dear Reviewer 5qFD,
>
> We thank you again for recognizing the contribution of our work to the RL/MARL community. We appreciate your time and hard work in providing valuable comments for our paper. Please let us know if you have any questions. More questions and discussions on our work are always welcome!
>
> Sincerely yours,
>
> Paper 2668 authors

---

### Official Review · Reviewer_9Jjr · 2022-10-26

**Confidence:** 4
**Correctness:** 3
**Technical Novelty And Significance:** 3
**Empirical Novelty And Significance:** 4
**Recommendation:** 3

**Clarity, Quality, Novelty And Reproducibility:**

As pointed out above in weakness, clarity has some room for improvement. Some further questions/comments I have include:
* There are many different (sub-)version of models. I think having a table to list them and also list the transfer experiments would be helpful.
* Why in Figure 4 both decoder and pooling outputs $z_i$?

Nitpick:
* Gammer error in "How the performance of the entity transformer in single game?" (beginning of Sec. 5)

**Strength And Weaknesses:**

Strengths
* It is a good proof of existence of generalist agents for multiple multi-agent environments. I think showing the community how to learn such agent is a valuable empirical contribution.
* The transfer-to-unseen-scenario results and transfer-to-new-game results are convincing, showing that training the pre-trained model with more games are usually beneficial.

Weaknesses
* I found Sec. 5.5 (scaling and comparison to single-game agents) confusing. With only scores normalized by single-agent performance at each size tick. I think it doesn't have enough information for me to interpret the results and the conclusion "We interpret this result as overparameterization, where a richer model is fitting than necessary" is kinda hand-wavy. I would suggest adding raw scores, which could help the reviewers to inspect the claim and help reader to understand better.
* The learning curriculum (Figure 5) is adhoc, and there is no ablation on the curriculum. The current curriculum basically uses Honor of Kings as an anchor. I think even just explaining what could happen otherwise would be helpful.

**Summary Of The Paper:**

This work proposes a learning framework, MAGENTA, that can tackle multiple multi-agent environments with one transformer agent. The authors focused on real-time strategy games in this work, including Honor of Kings (HoK), Starcraft II micromanagement (SMAC), and Neural MMO (NMMO). They treat agent entities as tokens and learn separated tokenizers, a unified transformer body, and separated output layers. It shows that it is possible to train generalist agents for multiple multi-agent environments in an online manner (using PPO), and shows that such online generalist agents can rapid adapt to never-seen-before games or scenarios with fine-tuning. The results empirically show that the proposed model learn common knowledge about multi-agent games across various categories.

**Summary Of The Review:**

I'm not able to interpret the results given the current manuscript, so couldn't recommend acceptance now. I hope the authors would be able to fix this.

---

> ### Author Response · Authors · 2022-11-14
> **Response to Reviewer 9Jjr**
>
> Thank you for carefully reviewing our paper! We greatly appreciate your feedback. Please see below our responses to your comments.
>
> -----------
>
> **Q1**: Sec 5.5
>
> **A1**: In Sec 5.5, we investigate the performance of different model sizes (parameters) in different games. Namely, we show a steady performance of M3 with 1,2,3 layers of transformer encoder, respectively. The score is normalized based on the performance of M1 with 1 layer of transformer encoder. We add the raw scores in the Appendix E.1 of the revised version. We interpret the result as overparameterization since we test a few-FC-layer model in SMAC and NMMO, and the performance is close to M1. The redundant parameters do not bring performance improvement. So we have this interpretation.
>
> -----------
>
> **Q2**: curriculum
>
> **A2**: We tested other curriculums, e.g., transferring from SMAC to HoK, however, the performance is not significantly different from that of M1 in HoK. Since the state and action space of HoK is much larger than those of SMAC/NMMO as shown in the Appendix. It again shows the overparameterization in SMAC/NMMO.
>
> -----------
>
> **Q3**: versions of models
>
> **A3**: We list the models and transfer experiments in Table 2 in the revised version.
>
> -----------
>
> **Q4**: Why in Figure 4 are both the decoder and pooling outputs $z$?
>
> **A4**: $z$ is a vector with a dimension of $d_v$. $d_v$ is the dimension of the value matrix in the attention module. Since the output of the encoder is $(n, d_v)$, where $n$ is the number of entities, the pooling merges along the n entities and the decoder uses $e_i$ as a query to extract information. Here we use $z_i$ to denote the permutation-invariant representation and do not differentiate $z_i$ from different models.
>
> -----------
>
> **Q5**: typos and grammar mistakes
>
> **A6**: We fix these issues in the revised version.

---

> > ### Comment · Reviewer_9Jjr · 2022-11-25
> > **Thank you for the response**
> >
> > Thanks for adding Table 2, though it is still quite confusing. In Sec 5.3, you say *$M^2$ is trained in HoK, $M^3_1$ in HoK, SMAC 5m_vs_6m scenario, NMMO 8 agent scenario, and $M^3_1$ in HoK, SMAC 8m_vs_9m, NMMO 8 agent scenario.*. It's quite confusing that you say $M^2$ is trained on 1 model $M^3$ is trained on 3 models.  I'm guessing what you mean is that you transfer $M^1_H$ to train $M^2_{S or N}$, and then transfer $M^3_1$ / $M^3_2$ to another SMAC scenario. Can you clarify? It seems like 5m_vs_6m and 8m_vs_9m are considered as scenarios instead of a separate environment. Why are they described in Sec 5.3 (Transfer to new games) instead of Sec. 5.4 (Transfer to new scenarios)? In Sec. 4.3, you said stage 3 ($M^3$) is for "Then we check whether M3 can handle never-seen-before scenarios in these games with few-shot transfer.", but the first ticks in Fig. 10 are already at 5 hours. Is this still few-shot? Same for $M^3$s in Fig 8, 9, they are transferred with up to one to a couple millions samples. Are these still considered few-shot? Despite the revision, it seems to me that the overall clarity of the paper is still quite poor.
> >
> > Please add what your response for the curriculum to the paper to clarify.
> >
> > Please differentiate $z_i$ in Fig. 4. Otherwise it's quite confusing.
> >
> > More typo in Sec 5.5: "We transfer different parts of MAGENTAto the new scenarios" missing a space

---

> > > ### Author Response · Authors · 2022-11-28
> > > **Re: Thank you for the response**
> > >
> > > Dear Reviewer 9Jjr,
> > >
> > > We appreciate your quick response to help us to improve this work. And thanks for pointing out the issue about curriculum and the clarity issue.
> > >
> > > ------------------------
> > >
> > > **Q1**: the confusing point about transferring models
> > >
> > > **A1**: Sorry for this confusing point. Here $M^i$ means models after the corresponding stage $i$. So $M^2$ means the model is trained in two games (trained in HoK and then transferred from HoK to SMAC/NMMO). and $M^3$ means the model is trained in three games (trained in HoK, SMAC, NMMO, and then transferred to new scenarios SMAC/NMMO). This is also what we want to express in Table. 2. We will clarify this in the revised version.
> > >
> > > -------------------------
> > >
> > > **Q2**: Sec 5.3 and Sec 5.4
> > >
> > > **A2**: 5m_vs_6m and 8m_vs_9m are two different scenarios in SMAC. In Sec 5.3 we describe Fig. 8 and in Sec 5.4, we analyze the result in Fig.8 by saying that “Then we look back at Fig.8 and Fig.9. We can see that $M^3$ can outperform $M^2$ in the held-out scenarios.” It corresponds to the statement in Sec. 4.3, “we check whether M3 can handle never-seen-before scenarios in these games with few-shot transfer”. We will revise this part to highlight this point.
> > >
> > > --------------------------
> > >
> > > **Q3**: Few-shot transfer
> > >
> > > **A3**: In few-shot transfer learning, the aim is to obtain models that can generalize from few samples. We think that the comparison between the transferred model trained with few samples and the model trained from scratch can be regarded as a few-shot transfer experiment. Figure 10 shows the transfer performance of $M^3$ during training in the new scenarios of HoK (trained in HoK 5v5 and transferred from 5v5 to 1v1). So it is a few-shot transfer since $M^3$ can learn faster with few-samples than models trained from scratch. And Figure 8 shows the transfer performance during training in the new games/scenarios. So it is also a few-shot transfer.
> > >
> > > ---------------------------------
> > >
> > > We summarized the points of this response:
> > >
> > > - We will add our response for the curriculum to the paper.
> > > - We will differentiate $z_i$ from different models by adding the superscript, e.g. $z^{ED}_i$,
> > > $z^{EP}_i$.
> > > - We will fix the typos
> > >
> > > Thanks again for your feedback. Since in the current stage of discussion, we authors cannot modify the paper. These points are clarity issues that are relatively easy to fix. We wish our response is helpful and we will further polish the paper to reduce the reading burden of the paper.
> > >
> > > Paper 2668 authors

---

> > > > ### Comment · Reviewer_9Jjr · 2022-12-04
> > > > **Suggestions**
> > > >
> > > > I understand the difference between $M^2$ and $M^3$. I strongly suggest the authors to rename these models. The fact that $M^2$ is pre-trained on 1 game but $M^3$ is pre-trained on 3 games is really confusing to readers. Also, as $M^3$ is for evaluating "transfer to new scenario", I would suggest not referring to it in Sec 5.3 at all to avoid confusion. For "few-shot" transfer, I suppose the authors know what the word "few" means. A million steps, not matter counted in number of steps or number of episodes, cannot be considered "few" if I understand correctly.

---

> > > > > ### Author Response · Authors · 2022-12-05
> > > > > **Re: Suggestions**
> > > > >
> > > > > Dear Reviewer 9Jjr,
> > > > >
> > > > > We really appreciate your valuable suggestions about this work. Based on your suggestions, we:
> > > > >
> > > > > - modify the models' names by using $M^{\text{expert}}, M^{\text{to new games}}, M^{\text{to new scenarios}}$ to replace $M^1, M^2, M^3$, and use subscript to describe the trained environments, e.g. $M_{H}^{\text{to new games}}$ and $M_{H,S,N}^{\text{to new scenarios}}$.
> > > > > - move the description of $M^{\text{to new scenarios}}$ from Sec 5.3 to Sec 5.4.
> > > > > - rename "few-shot transfer" to "transfer".
> > > > >
> > > > > As for the "few-show transfer", according to [1], "Few-shot Learning (FSL) is a type of machine learning problems (specified by E, T, and P), where E contains only a limited number of examples with supervised information for the target T." Since the RL problems require a huge number of data, even though the transfer in RL needs much less data than training data (about 30%-50% of training data in our experiments), we agree that millions of steps cannot be considered as "few". So we rename "few-shot transfer" to "transfer"
> > > > >
> > > > > Thanks again for your suggestions about this work. We hope our response can address your concerns.
> > > > >
> > > > > [1] Generalizing from a Few Examples: A Survey on Few-shot Learning. 2020.
> > > > >
> > > > > Paper 2668 authors

---

> ### Author Response · Authors · 2022-11-25
> **Follow-up Response**
>
> Dear Reviewer 9Jjr,
>
> We thank you again for recognizing the contribution of our work to the RL/MARL community. We appreciate your time and hard work in providing valuable comments for our paper. Please let us know if you have any questions. More questions and discussions on our work are always welcome!
>
> Sincerely yours,
>
> Paper 2668 authors

---

### Author Response · Authors · 2022-11-14
**General response to all reviewers and summary of the revised version**

We thank all reviewers for carefully reviewing our paper! We greatly appreciate your feedback. Please see below our general responses and summary of the revised version.

------------------

**Q1**: Novelty

**A1**: Our aim is to build generalist agents for multiple multi-agent environments. Even though the methodological backbone of MAGENTA is PPO+transformer, which is not novel in the RL community. We provide a new perspective of analogizing games as languages and propose novel architectures for permutation-invariance.

-----------------

**Q2**: Reproducibility

**A2**: We uploaded our source code in the supplementary materials. We will opensource our code and model parameters after the double-blind review process.

-----------------

**Q3**: The clarity of the paper

**A3**: We revised our paper and details can be seen below.

-----------------

Summary of the revised version (We use blue color to highlight the main changes):

- Modify the abstract and introduction to show our perspective of viewing games as languages. (Q1 of Reviewer gmYJ)
- Move related work about RL in games to the appendix. (Minor comments in Reviewer gmYJ)
- Revise Sec 4.2 and Appendix F about the details of transformer in MAGENTA. (Q4 of Reviewer 9Jjr and Q1 of Reviewer D4zs)
- Add table 2 in Sec 5.1 about the the models and transfer experiments. (Q3 of Reviewer 9Jjr)
- Other fixes about typos, grammar, and figures.

---

> ### Author Response · Authors · 2022-11-16
> **More revisions**
>
> Based on Reviewer gmYJ's suggestion, we:
>
> - modify the statement in the Introductioon by saying: First, when the training environment and the test environment are the same, RL is inclined to overfit the training environments, while lacking generalizability to unknown environments.
>
> - add the examples of images and vectors in the Sec 4.1 and Appendix.
>
> We thanks all reviewers' suggestion and constructive comments to help us to improve this work. If you have any further questions or comments, please post them and we will be happy to have further discussions.

---

> > ### Author Response · Authors · 2022-11-18
> > **Follow up revision**
> >
> > Dear All reviewers,
> >
> > We hope that the revised version is helpful in improving your rating. Please let us know if you have any other comments or feedback. We will be happy to incorporate further feedback in the final version of the paper. We are looking forward to hearing back from you!
> >
> > Thank you!
> >
> > Paper 2668 authors

---

> > > ### Author Response · Authors · 2022-12-07
> > > **More revisions: Part 2**
> > >
> > > Based on Reviewer 9Jjr's suggestions, we:
> > >
> > > - modify the models' names by using $M^{\text{expert}}, M^{\text{to new games}}, M^{\text{to new scenarios}}$ to replace $M^1, M^2, M^3$, and use subscript to describe the trained environments, e.g. $M_{H}^{\text{to new games}}$ and $M_{H,S,N}^{\text{to new scenarios}}$.
> > > - differentiate $z_i$ from different models by adding the superscript, e.g. $z_i^{EP}$, and $z_i^{ED}$.
> > > - move the description of $M^{\text{to new scenarios}}$ from Sec 5.3 to Sec 5.4.
> > > - rename "few-shot transfer" to "transfer".
> > >
> > > Since in the current stage of discussion, we authors cannot upload the modified paper. These points are clarity issues that are relatively easy to fix. We wish our response is helpful and we will further polish the paper to reduce the reading burden of the paper.

---

### Decision · Program_Chairs · 2023-01-20

**Decision:**

Reject

**Justification For Why Not Higher Score:**

Too many unresolved issues with clarity on what experiments were run

**Justification For Why Not Lower Score:**

N/A

**Metareview: Summary, Strengths And Weaknesses:**

This paper explores the application of pre-trained transformers to decision making in multi-agent systems. Conceptually, this could be a significant step forwards from recent work applying these models in single agent games. However, the reviewers raised many issues with the clarity of the paper initially submitted. The authors have made good progress on addressing the minor issues during the discussion period, but there remains large uncertainty over the precise model architecture and experiments reported on. Without these major issues addressed the insights that can currently be gained by an external reader are limited.